# Enhancing Human Body Generation in Diffusion Models with Dual-Level Prior Knowledge

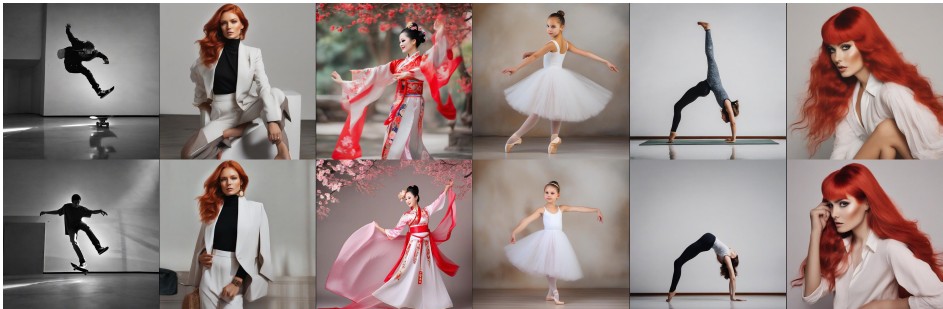

Figure 1: Comparison between images generated by pretrained SDXL (first row) and SDXL fine-tuned with our method (second row).

## ABSTRACT

The development of diffusion models (DMs) has greatly enhanced text-to-image generation, outperforming previous methods like generative adversarial networks (GANs) in terms of image quality and text alignment. However, accurately generating human body images remains challenging, often resulting in disproportionate figures and anatomical errors, which limits their practical applications in areas such as portrait generation. While previous methods such as HcP have shown promising results, limitations including retention of noisy priors, limited understanding of human representation, and restriction of generalization power, still exist due to the specific design of fully-supervised learning with only pose-related information. In this study, we introduce a novel method to enhance pretrained diffusion models for realistic human body generation by incorporating dual-level human prior knowledge. Our approach involves learning shape-level details with the human-related tokens in the original prompts, and learning pose-level prior by adding a learnable pose-aware token to each text prompt. We use a two-stage training strategy to rectify the cross attentions with a bind-then-generalize process, leveraging multiple novel objectives along with adversarial training. Our extensive experiments show that this method significantly improves the ability of SD1.5 and SDXL pretrained models to generate human bodies, reducing deformities and enhancing practical utility.

## 1 INTRODUCTION

Despite improvements in AI-generated content fidelity and text alignment compared to GANs, diffusion models still struggle with human body images, a common visual in real-world media. These advanced models often produce disproportionate bodies and inaccurately placed, missing, or redundant limbs and hands, limiting their application in areas like portrait generation and image stylization. Since the introduction of the Latent Diffusion Model (LDM) Rombach et al. (2022), researchers have focused on scaling up model and training data sizes to enhance human body image quality. However, even SDXL in Fig. 1 and the recent SD3 Esser et al. (2024) continue to strug-

gle with this issue, suggesting that simply increasing model or data size does not ensure effective representation of human body shapes.

We recognize HcP Wang et al. (2024a) as one of the few recent studies focused on improving human body generation. This work introduces an additional attention map derived from pose information to amend the original cross-attention mechanisms. Unlike other methods Liu et al. (2023) that rely on explicit conditions such as pose or depth, HcP can be seamlessly integrated into any text-to-image diffusion model, making it highly valuable. The study highlights a significant issue: *the cross-attention modules in diffusion models often struggle to capture the location and shape information of target objects related to human tokens*. This challenge is illustrated in Fig. 2(a), where examples generated by the pretrained SDXL show that cross-attention maps for human-related tokens either marginally highlight or completely neglect human regions, resulting in noisy, large activations in the background.

While HcP delivers impressive results and maintains a lightweight design, we still identify several issues existed in this approach: 1) **Retention of Noisy Priors**: HcP revises the cross-attention maps of pretrained diffusion models by introducing additional attention maps. This method retains all prior knowledge from the pretrained models, including noisy attention maps. The new branch can highlight the desired regions but cannot suppress other areas, resulting in continued noisy attention maps. 2) **Limited Understanding of Human Representation**: HcP primarily focuses on human poses, neglecting other essential aspects of human representation. A pose alone cannot specify a particular human body type. Factors such as body shape (e.g., thin or heavy) and occlusion by objects (as shown in Fig. 2(c)) are also crucial for accurately depicting human figures. For instance, with the pose illustrated in Fig. 2(b), it remains unclear whether to generate a slender girl or a heavier one. 3) **Restriction of Generalization Power**: HcP adopts a fully-supervised learning strategy, relying on pose annotations paired with each training image. However, real-world images often feature diverse human bodies in unexpected poses. Even well-trained segmenters like SAM Kirillov et al. (2023) and pose detectors such as HrFormer Yuan et al. (2021) can struggle in various corner cases. As a result, while the pretrained diffusion models may perform well on the training data, their generalization ability is limited.

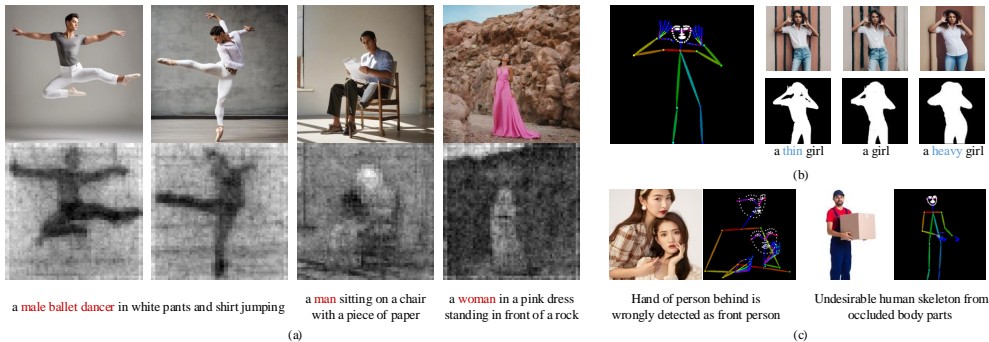

Figure 2: (a) Visualization of images generated by pretrained SDXL and their cross attention maps with regard to human-related tokens which are marked as red in prompts. For the attention maps, white color stands for large activation. The pretrained SDXL fails to produce meaningful cross attention maps for human-related tokens. (b) With the same pose, humans with different shapes can be generated, which means more human body prior besides pose information should be leveraged for better human body generation. (c) Pose detected by deep models can be very noisy in some cases.

To address the challenges in generating realistic human bodies, we propose a novel dual-level prior injection method that enhances pretrained diffusion models by learning both shape-level and pose-level human priors. Unlike HcP, which captures only limited human priors, our approach aims for a more generalizable and comprehensive understanding through a two-stage bind-then-generalize strategy. Specifically, we first bind the human-related tokens from the original prompts, along with an additional pose token attached to the text prompt, to their corresponding semantic regions. This binding produces meaningful attention maps for these tokens. We propose a novel composite objective with three distinct terms to facilitate the learning process: The first term suppresses attention

activation outside human regions in each image. Building on this, the second term encourages the diffusion UNet to ensure that human-related activation exceeds non-human-related activation, effectively highlighting human body regions. The third term serves as an auxiliary objective to learn the scale of the ground truth. Together, these objectives empower the pretrained diffusion models to extract and aggregate critical information from various token types, leading to higher-quality human body generation.

After establishing a one-to-one mapping based on the training data, we guide the model to mimic the distribution of real attention maps. We utilize commonly-used Generative Adversarial Networks (GANs) by introducing shape-aware and pose-aware discriminators corresponding to the human-related and pose tokens, respectively. The diffusion UNet is trained against these discriminators in an adversarial minimax game, enabling it to generate cross-attention maps that align with real data distributions. This process ultimately enhances the model's ability to produce realistic human bodies across various contexts, including diverse real-world scenarios.

To show the effectiveness of our proposed method, we conduct extensive experiments across various text prompts. Our findings reveal that with simple tuning, both the pretrained SD1.5 and SDXL significantly enhance their capacity to generate human bodies, resulting in fewer problematic and deformed outputs, thus showing greater practical value. In summary, the contributions of this work are as follows:

1)*Dual-Level Prior Injection Method*: We present a novel method that enhances pretrained diffusion models by learning shape-level and pose-level human priors through a two-stage bind-then-generalize strategy. Our method binds human-related tokens from text prompts and an additional pose token to their corresponding semantic regions, resulting in meaningful attention maps.

2)*Composite Objective for Learning*: We propose a new composite objective that comprises three novel terms: one for suppressing noisy activations, another for enhancing human activation, and a third for learning the scale of ground truth.

3)*Mimicking Real Attention Map Distribution*: We further introduce adversarial training to rectify the cross attention maps, with shape-aware and pose-aware discriminators enhancing the model's ability to mimic the real data distribution.

4)*Improved Model Performance*: Extensive experiments demonstrate that simple tuning can significantly improve the performance of pretrained models (SD1.5 and SDXL) in generating human bodies, resulting in fewer deformed outputs and showcasing greater practical value.

## 2  RELATED WORK

**Refined human body generation.**   In spite of astonishing image fidelity in general, the diffusion models have long been suffering from problems of inaccurate details, especially human bodies. To solve this problem and generate better human images, two main strategies are adopted. The first is extra conditions. For example, ControlNet Zhang et al. (2023) was proposed to add additional spatial controlling signal to the denoising process through a replicated branch of diffusion UNet encoder. Following this work, many other works, such as HyperHuman Liu et al. (2023), tried to apply more comprehensive conditions including depth and normal maps to diffusion models for better results. Huang et al. Huang et al. (2024) proposed to generate human bodies in a hierarchical manner, with parts first being generated and then whole bodies. Another important strategy is regularization. HcP Wang et al. (2024a) tried to learn an additional attention branch with pose-related information, thus making diffusion models aware of the poses and generating more reasonable human bodies. Our paper follows the idea of regularization. In comparison, after training with our proposed method, diffusion models can be used in text-to-image generation as normal, without extra process. Furthermore, compared with HcP, we design a more refined pipeline in order to inject the prior knowledge regarding human body into pretrained diffusion models, including adopting a two-stage training regime and leveraging adversarial training, which has not been considered in the previous works.

**Adversarial training for diffusion models.**   Adversarial training was originated in Generative Adversarial Models (GANs) Goodfellow et al. (2020) that learn to mimic the real data distribution as competing procedure between generator and discriminator, which iteratively improve both models' performance. Recently, some researches have been focusing on enhancing diffusion models with adversarial training. Most of them are aimed at reducing the sampling steps of DMs. Xiao et.

al. Xiao et al. (2021) proposed to replace the minimization of divergence with Gaussian distribution with the adversarial divergence with a learned discriminator. Xu et al. Xu et al. (2024) further improved this formulation with simpler objective and better training strategy, thus helping diffusion models generate high-quality images with fewer sampling steps. ADD Sauer et al. (2023), following the same idea, proposed to combine the distillation and adversarial training to further enhance the efficiency of diffusion models. On the other hand, adversarial training is also applied to diffusion models for better generation quality. For example, Li et al. Li et al. (2024) proposed to discriminate the noisy estimation of generated images with a segmenter. As the result, the diffusion models can better follow the control of input segmentation masks. Besides, Yang et al. Yang et al. (2024) proposed to leverage adversarial training to embed structural prior into diffusion models. Our method follows the idea of improving performance of diffusion models with adversarial training. However, different from the previous works that rely on the noisy image estimation, we propose to apply discriminator to the intermediate cross attention maps, which leads to significantly more efficiency while guaranteeing strong performance.

## 3 PRELIMINARY: STABLE DIFFUSION

Diffusion models aim to capture the data distribution $p_\theta(\mathbf{x}_0)$ of clean data $\mathbf{x}_0$ by gradually refining a standard Gaussian distribution, with the learning process framed as denoising score matching. Stable Diffusion (SD) builds upon this framework to enable text-to-image generation based on a text prompt $p$. By leveraging a pre-trained VQ-VAE Van Den Oord et al. (2017) that includes an encoder $\mathcal{E}$ and a decoder $\mathcal{D}$, SD allows the model to concentrate more on the semantic aspects of the data, thereby enhancing efficiency. A diffusion UNet is utilized to estimate the noise, incorporating an attention mechanism. Specifically, in the $l$-th layer, self-attention is first employed to facilitate interactions among spatial features: $z^l = Attention(W_Q^l \cdot z^l, W_K^l \cdot z^l, W_V^l \cdot z^l)$, where $Attention$ denotes the attention operator, $z^l$ represents the latent embeddings of the $l$-th layer, and $W_Q, W_K, W_V$ are the projection layers for self-attention. Following this, cross-attention is applied to incorporate conditioning information such as the text prompt: $\hat{z}^l = Attention(\hat{W}_{Q_t}^l \cdot z^l, \hat{W}_{K_t}^l \cdot z_{text}, \hat{W}_{V_t}^l \cdot z_{text})$, where $z_{text}$ denotes the text prompt embedding, and $\hat{W}_Q, \hat{W}_K, \hat{W}_V$ are the projection layers for cross-attention. The training objective of SD is formulated as follows:

$$\mathcal{L}_{noise} = \mathbb{E}_{\mathcal{E}(x), \epsilon \sim \mathcal{N}(0,1), t} \left[ \left\| \epsilon - \epsilon_\theta\left(z^t, t\right) \right\|_2^2 \right], \tag{1}$$

where $t$ is uniformly sampled from $\{0, ..., T\}$, and $z^t$ represents the noisy latent at the $t$-th timestep.

## 4 METHODOLOGY

### 4.1 SEMANTIC ATTENTION BINDING

Previous works have shown that the cross attention maps of diffusion UNet between text prompts and image latent embeddings can indicate the coarse location and shape of the target objects. However, as mentioned in Sec. 1, the pretrained SD fail to produce meaningful cross attention maps regarding those human-related token, such as man, woman, etc, which can result in the problem of human body deformity. Consequently, we propose to first guide the cross attention maps to bind with correct regions, i.e., suppressing non-human-related activation and highlighting human-relation one. Inspired by previous works, a composite objective is utilized which including the following three items.

**Suppressing loss.** To serve our goal, we first utilize a simple objective to suppress the attention activation outside the human bodies. Particularly, when processing each noisy latent $z^t$ corresponding to image $\mathbf{I}$ with the diffusion UNet, we collect all intermediate cross attention maps and upsample them to fixed size, e.g., $512 \times 512$, denoted as $\hat{\mathcal{M}}^{shape} \in \mathbb{R}^{512 \times 512}$. In the mean time, SAM Kirillov et al. (2023) is leveraged to segment all human bodies in $\mathbf{I}$, resulting in a shape mask $M^{shape}$. Then the suppressing loss is calculated as follow:

$$\mathcal{L}_s^{shape} = \sum_{\hat{M} \in \hat{\mathcal{M}}^{shape}} \sum_{i,j} \hat{M}_{i,j} \mathbb{1}(M_{i,j}^{shape} = 0) \tag{2}$$

where $i, j$ denote spatial index, $\mathbb{1}$ denotes the indicator function.

**Margin loss.** In order to teach the diffusion UNet to highlight human-relation activation, a margin-based objective is utilized as follows:

$$f(\hat{M}) = \min_{i,j}(\hat{M}_{i,j} \mathbb{1}(M_{i,j}^{shape} = 1)) \qquad (3)$$

$$g(\hat{M}) = \max_{i,j}(\hat{M}_{i,j} \mathbb{1}(M_{i,j}^{shape} = 0)) \qquad (4)$$

$$\mathcal{L}_{margin}^{shape} = \sum_{\hat{M} \in \hat{\mathcal{M}}^{shape}} \max(0, f(\hat{M}) - g(\hat{M}) + \delta) \qquad (5)$$

where $\delta$ denotes the margin coefficient. By leveraging this loss, the model is encouraged to produce attention maps in which human-related activation is at least larger than non-human-related activation, thus better highlighting the human body regions in each image.

**Scaling loss.** The above two objectives can encourage the model to produce attention maps that have large values inside human bodies and small values outside. To further enhance the training, we utilize a scaling loss to directly let the attention maps recover the scale of $M^{shape}$.

$$\mathcal{L}_{scale}^{shape} = \frac{1}{HW} \sum_{\hat{M} \in \hat{\mathcal{M}}^{shape}} \|\hat{M} - M^{shape}\|_2^2 \qquad (6)$$

By training the diffusion UNet with the combined loss

$$\mathcal{L}_{bind}^{shape} = \mathcal{L}_s^{shape} + \mathcal{L}_{margin}^{shape} + \mathcal{L}_{scale}^{shape}, \qquad (7)$$

the cross attention modules can learn what information is represented by the human-related tokens, and aggregate these information into right regions in each image, thus sketching reasonable human shapes. TokenCompose Wang et al. (2024b) also proposes to regularize cross attention maps of diffusion UNet with extra objectives. However, since TokenCompose focuses on improving the textual fidelity of diffusion models, it is sufficient for the cross attentions to provide rough locations for each semantic token. Compared with that, rectifying human body deformity requires more refined attention control, otherwise the unrelated information could easily lead to mistakenly drawn human bodies. Therefore, we propose a composite objective with multiple sub-goals which can provide stronger supervision as we will show in the experiments.

**Pose tokens.** The proposed $\mathcal{L}_{bind}^{shape}$ can guide the diffusion UNet to attend the human-related tokens in text prompts on human-related regions in the images. However, it is hard to correctly portray a human body solely based on the shape information. For example, when a man crosses his arms, $M^{shape}$ can only show that the arms are not stretched, but cannot tell what exact pose the man is holding. As the result, it is also important to embed to pose-related information into the diffusion UNet. In spite of its importance, simply copying $\mathcal{L}_{bind}^{shape}$ to train the human-related tokens also with pose information means these tokens have to learn two different levels of knowledge, which would be much harder.

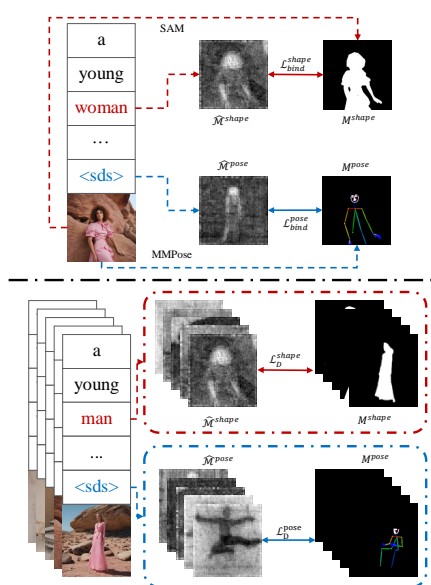

Figure 3: Schematic diagram of our proposed method. Up: The first stage utilizes fully-supervised learning to learn both shape and pose priors for human-related tokens and extra pose token respectively. Bottom: Instead of solely leveraging fully-supervised learning, we adopt unsupervised adversarial training to mimic the distribution of real shapes and poses, thus entitling the model with better generalization ability.

To this end, we propose a simple yet effective alternative method. Specifically, a learnable pose token $<sds>$, which is aimed to represent pose-related information, is attached to each text prompt.

To supervise the pose token with prior knowledge regarding poses, for each image $\mathbf{I}$, we leverage MMPose Contributors (2020) to detect the keypoints $\{p_i\}_{i=1}^{N_p}$, where $N_p$ denotes the maximum number of keypoints. Then $\{p_i\}$ is processed into a skeleton mask $M^{pose}$, in which lines are drawn between adjacent keypoints following human structure, *e.g.*, neck and shoulders, elbows and wrists. After setting up $<sds>$ and $M^{pose}$, we can build the training scheme following the same way as guiding human-related tokens. Concretely, for each training iteration, we collect the cross attention maps $\hat{\mathcal{M}}^{pose}$ regarding $<sds>$, and calculate the following objectives:

$$\mathcal{L}_{bind}^{pose} = \mathcal{L}_s^{pose} + \mathcal{L}_{margin}^{pose} + \mathcal{L}_{scale}^{pose} \tag{8}$$

in which each item follows the same formulation as in Eq. 7, with $\hat{\mathcal{M}}^{shape}, M^{shape}$ replaced with $\hat{\mathcal{M}}^{pose}, M^{pose}$. To make full usage of the proposed objectives, we attach additional LoRA parameters Hu et al. (2021) to pretrained SD models, and optimize them together with the newly added token embedding corresponding to pose token $<sds>$ with $\mathcal{L} = \mathcal{L}_{noise} + \mathcal{L}_{bind}^{shape} + \mathcal{L}_{bind}^{pose}$.

## 4.2 ADVERSARIAL ATTENTION RECTIFICATION

While the above training strategy can guide the model to correctly bind human-related tokens with their semantic regions in the training set, such property can hardly be generalized to unseen cases, since the shape and pose of humans can severely vary among different images according to their ages, genders, events and even the background scenarios. This makes the distribution of shape and pose data a sparse one, leading to great difficulty for the model to transfer the learned knowledge to wide range of real cases. To solve this problem, we further propose to leverage adversarial training which directly learns the data distribution via the supervision of a learnable discriminator rather than reciting the one-to-one mapping.

Specifically, we setup two discriminators $D^{shape}, D^{pose}$ for shape and pose data respectively, corresponding to the human-related tokens and extra pose tokens. During each training iteration, we first get the intermediate cross attention map sets $\hat{\mathcal{M}}^{shape}, \hat{\mathcal{M}}^{pose}$, together with their ground truth annotations $M^{shape}, M^{pose}$, using the same way as in Sec. 4.1. Based on these data the adversarial training can be built to bridge the gap between the cross attention maps predicted by diffusion UNet, *i.e.*, fake distribution, and the shape/pose masks of real images, *i.e.*, real distribution. We simply follow WGAN-GP Gulrajani et al. (2017) for the instantiation. Formally, the discriminators are optimized by minimizing the following objectives:

$$\mathcal{L}_D^{shape} = \frac{1}{|\hat{\mathcal{M}}^{shape}|} \sum_{\hat{M} \in \hat{\mathcal{M}}^{shape}} D^{shape}(\hat{M}) - D^{shape}(M^{shape}) + \alpha \Delta^{shape} \tag{9}$$

$$\mathcal{L}_D^{pose} = \frac{1}{|\hat{\mathcal{M}}^{pose}|} \sum_{\hat{M} \in \hat{\mathcal{M}}^{pose}} D^{pose}(\hat{M}) - D^{pose}(M^{pose}) + \alpha \Delta^{pose} \tag{10}$$

where $\Delta$ denotes the gradient penalty, $\alpha$ denotes the coefficient. Accordingly, the diffusion UNet is optimized as follows:

$$\mathcal{L}_U = \mathcal{L}_{noise} + \mathcal{L}_G^{shape} + \mathcal{L}_G^{pose} \tag{11}$$

$$\mathcal{L}_G^{shape} = -\frac{1}{|\hat{\mathcal{M}}^{shape}|} \sum_{\hat{M} \in \hat{\mathcal{M}}^{shape}} D^{shape}(\hat{M}) \tag{12}$$

$$\mathcal{L}_G^{pose} = -\frac{1}{|\hat{\mathcal{M}}^{pose}|} \sum_{\hat{M} \in \hat{\mathcal{M}}^{pose}} D^{pose}(\hat{M}) \tag{13}$$

Through the adversarial minimax game, the diffusion UNet can gradually learn how to adapt the LoRA weights so that the cross attention maps can be rectified to ideal shapes and poses. Compared with previous works Li et al. (2024) that also adopt adversarial supervision for diffusion models, the most noticeable difference is that our method does not rely on the noisy prediction $\hat{z}^0$ achieved from $\hat{z}^t$, which results in two merits. First, the VAE decoder is not required for calculating discriminator prediction, thus being more efficient. Second, the input of discriminators are binary masks which are much easier than RGB images, thus making them better functioning for producing guidance supervision for the diffusion UNet.

## 5 EXPERIMENTS

### 5.1 IMPLEMENTATION DETAILS

**Dataset.** To train our model, we collect a specific dataset via crawling from open-source search engines and filtering out all images containing unclear human bodies, which results in a training set with 176,092 images. Then BLIP2 Li et al. (2023) is utilized to provide the captions corresponding to these data. As for validation, we follow HcP Wang et al. (2024a) to adopt the captions from validation set of HumanArt Ju et al. (2023) in the quantitative comparison. Besides, for qualitative results, we also include 50 prompts that are manually created with complex semantic meaning, on which we empirically find that the pretrained SD tends to generate deformed human bodies.

### 5.2 QUANTITATIVE RESULTS

Table 1: Quantitative result with SDXL as backbone. For FID and KID, the smaller score denotes the better model. For other metrics, the larger score denotes the better model.

| Methods | FID ↓ | KID ↓ | CLIP ↑ | HPS ↑ | PickScore ↑ |
|---------|-------|-------|--------|-------|-------------|
| Pretrain | 41.87 | 12.44 | 33.94 | 23.06 | 22.38 |
| LoRA | 40.34 | 12.54 | 34.68 | 23.06 | 22.31 |
| Ours | **38.11** | **11.77** | **34.84** | **23.17** | **22.61** |

We first present quantitative evaluations in this part. Concretely, FID Heusel et al. (2017), KID Bińkowski et al. (2018), CLIP-Score, HPS-v2 Wu et al. (2023) and PickScore Kirstain et al. (2023) are calculated to evaluate the general quality and textual fidelity of generated images. Given that previous methods such as HcP and HyperHuman are not open-sourced, we compare our method with pretrained SDs and LoRA finetuned model using our training set. The results are shown in Tab. 1. We find that directly finetuning pretrained SDXL makes most of the metrics better, and adopting our proposed strategy leads to further better results, which indicates the efficacy of our method. However, while we have presented quantitative metrics among various aspects, they generally concern about the image quality and textual fidelity rather than evaluating the quality of human body generation, and there lacks a fully related metric for our problem. Therefore, we advocate focusing more on the following qualitative results.

### 5.3 QUALITATIVE RESULTS

We present several uncurated results for SDXL in Fig. 4. For each prompt, we randomly select three random seeds to control the generation process of different methods, thus making sure the results can be fairly compared. Interestingly, we find that while the prompts are generally short and easy for human to understand, they are especially hard for pretrained SDXL to generate corresponding images. For example, when generating common actions such as sitting or jumping, which could be abundant in the pretrain dataset, the pretrained SDXL still cannot produce satisfactory results, leaving many artifacts such as unorganized body, redundant arms and hands, and improperly shaped limbs. When it comes to rare actions (*e.g.*, doing yoga, dancing ballet) or rare combinations of actions and objects (*e.g.*, holding a rifle), the deformity becomes more severe. As our baseline method, we find that finetuning SDXL on our dataset directly with LoRA does learn some information from the data, given that the finetuned model can generate much detailed background in many images. Unfortunately, this straightforward method has no positive effect and even makes it worse in many cases with regard to human body generation. For example, when generating according to the prompt '*a woman doing a yoga pose on a yoga mat*', the LoRA-finetuned SDXL can hardly generate a complete human body. This indicates that even if models with sufficient capacity (*e.g.*, SDXL) can be finetuned with specific training data (*e.g.*, high-quality human images), the problem of human body deformity can still not be solved. Compared with these methods, our method can guide the model to learn appropriate human body prior from the training data, resulting in much better human bodies. Meanwhile, our method does not require additional design for the network structure and input conditions, thus making the best of both worlds.

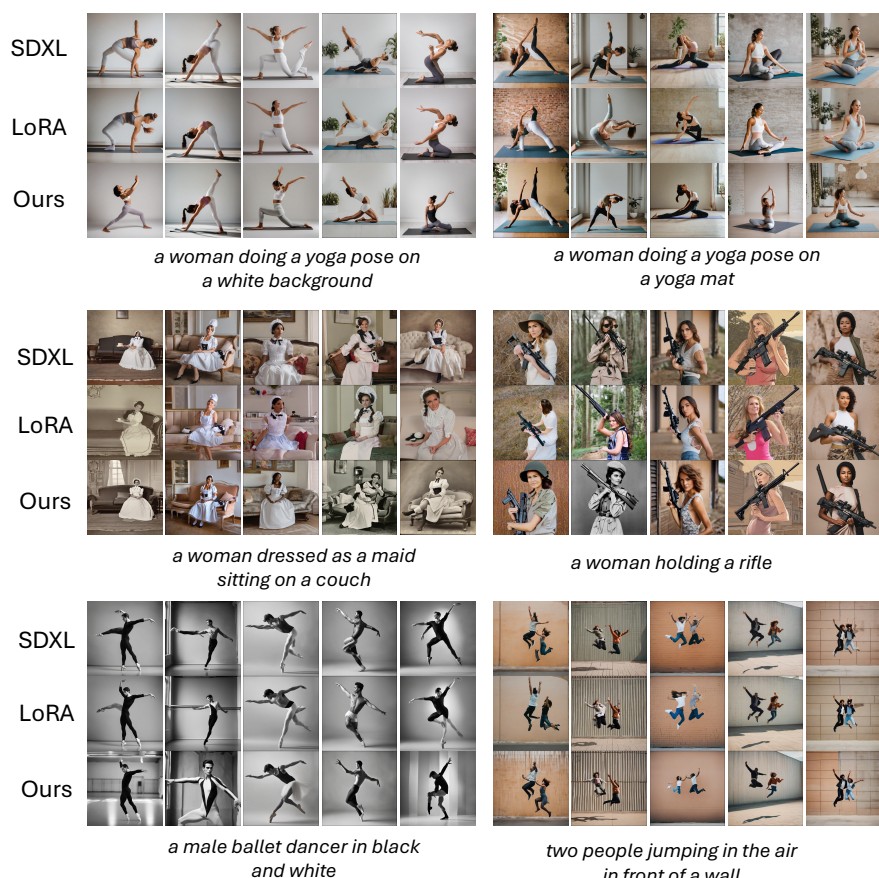

Figure 4: Qualitative comparison with SDXL as backbone. Each comparison is controlled with the same random seed.

To further show the generalization ability of our method, we apply it to pretrained SD1.5, whose results are presented in Fig. 5. The results are consistent with those of SDXL. In general, since SD1.5 itself is worse than SDXL, the image quality among three methods lags behind images generated by SDXL in terms of the details. Nonetheless, our proposed method still can help the model fix the problem of human body deformity, leading to more reasonable human bodies under different kinds of prompts.

## 5.4 ABLATION STUDY

To further verify the effectiveness of our proposed method, we conduct extensive ablation studies using SDXL as backbone. Since the above adopted quantitative metrics cannot intuitively reflect the quality of generated human bodies, we only provide the qualitative results, with parts of them shown in the main paper. For more comprehensive comparison please refer to the appendix.

**How can $\mathcal{L}_{bind}$ help the model?** We first test the efficacy of each item in $\mathcal{L}_{bind}$ as illustrated in Eq. 7, of which the generated images together with the averaged cross attention maps regarding human-related tokens are shown in Fig. 6. Same as the results in Sec. 1, we can find that the pretrained SDXL cannot concentrate on the generated human body regarding the human-related tokens. In some cases (*e.g.*, the second and fourth from left) the model even highlights more on background than human bodies. Consequently it cannot gather correct information, thus leading to deformed human bodies. Directly finetuning SDXL with LoRA also does not work, resulting in almost the same attention maps as the pretrained model. This indicates that the noise prediction loss $\mathcal{L}_{noise}$ cannot solely guide the cross attention modules to function correctly, hence leveraging the additional objectives is necessary. As for the three items proposed by us, it can be found that when using only

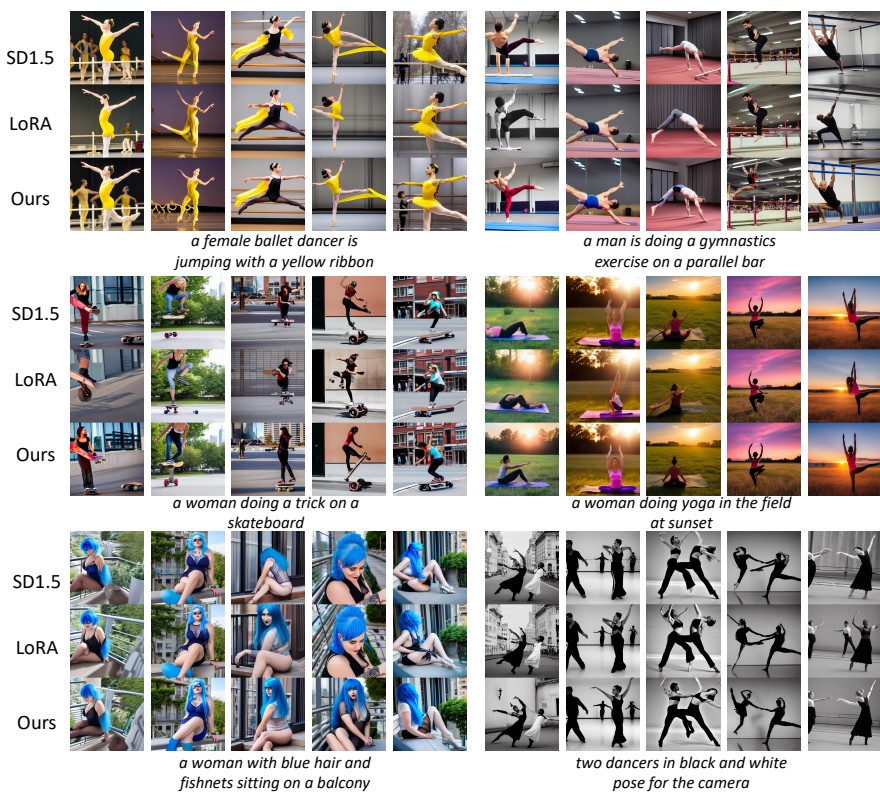

Figure 5: Qualitative comparison with SD1.5 as backbone. Each comparison is controlled with the same random seed.

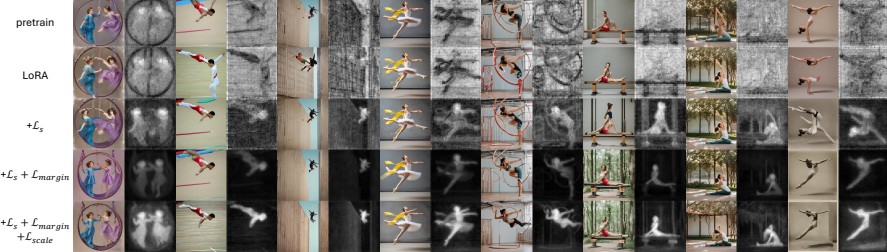

Figure 6: Ablation comparison among model variants using different objectives.

suppressing loss, the attention maps are better, but still noisy. When using both suppressing loss and margin loss, the model can fully concentrate on human bodies, but the activation scale is smaller. Introducing the scaling loss help the model highlighting human bodies with large activation values, thus leading to the best results.

**Effectiveness of the pose token.** In Fig. 7 we present three model variants regarding the learning strategy for human body prior: (1) **OnlyShape**: We only learn shape-related information with original human-related tokens in the prompts. (2) **OnlyPose**: Similar to OnlyShape, with shape-related information replaced with pose-related on. (3) **Ours**: Our proposed strategy as in Sec. 4.1, *i.e.*, using extra pose token to learn pose information. To make the results simple for understanding, we average attention activation among all human-related tokens and pose token. Basically, solely learning shape-level information can help the model better concentrate on the human bodies. However, since pose information is not injected, the model cannot avoid problem of abnormal poses, *i.e.*, missing or additional limbs. On the other hand, it is also difficult for human-related tokens to learn pose information, considering the results that the attention maps can only concentrate on specific regions such as faces. Compared with these two variants, our proposed strategy can help the model leveraging

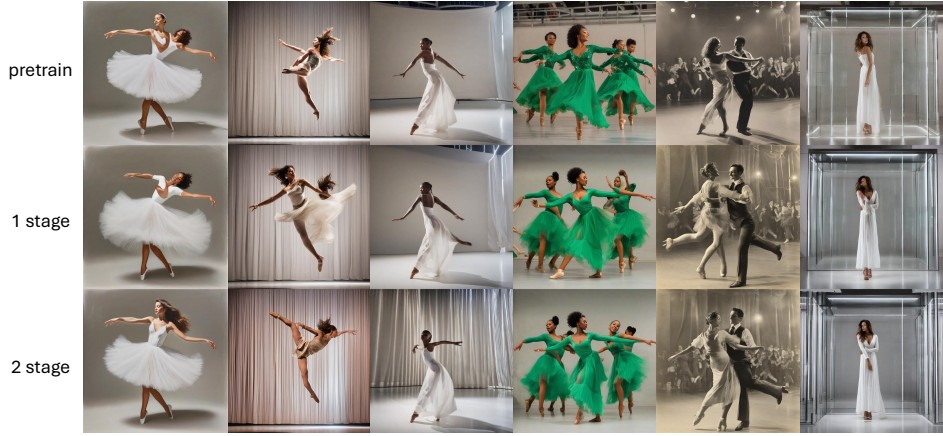

Figure 7: Ablation comparison among model variants using different training strategies.

attention maps that truly highlight the reasonable human body regions, thus generating human body images with higher quality.

**Effectiveness of adversarial training.** In Fig. 8 we compare three model variants including pretrained SDXL, SDXL finetuned with only the semantic attention binding stage proposed in Sec. 4.1 and the one finetuned with both semantic attention binding and adversarial attention rectification. We can find that the first stage can help the model generate better human bodies in some cases. For example, in the first column, while the pretrained SDXL generates an additional head, the 1-stage model can revise this mistake. However, the human bodies generated by this 1-stage model are often stilted, while also being prone to deformity, which shows its limited generalization ability due to fully supervised learning. In comparison, the model trained with both stages as shown in the last row can be generalized to different prompts, resulting in both more appropriate human poses and shapes.

Figure 8: Ablation comparison among model variants using different training strategies.

## 6 CONCLUSION

This paper introduces a novel method to enhance pretrained diffusion models for generating realistic human body images. By incorporating dual-level human body prior knowledge through a learnable pose token and human-related tokens, our approach addresses common issues like disproportionate figures and anatomical inaccuracies. Our two-stage training process, which includes binding tokens to semantic regions and leveraging adversarial training, significantly improves the fidelity and accuracy of generated human images. Experimental results demonstrate that our method effectively reduces deformities and enhances the practical utility of diffusion models, as shown by the improved performance of both pretrained SD1.5 and SDXL. This advancement surpasses previous approaches and opens new possibilities for real-life applications.

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

## A TRAINING DETAILS

To validate the efficacy of our method, we conduct experiments on pretrained SD1.5 and SDXL. We adopt AdamW as optimizer with 5e-6 learning rate. Our model is trained for 10,000 iterations for each stage on 16 V100s with 4 batch size on each gpu, which takes about 2 days. For the second stage, we adopt a 11-layer convolutional discriminator, which can produce patch-level predictions for each input attention maps. We empirically find that the discriminator could easily be too strong to fool, making the adversarial training less effective. To this end, dropout layers are added to the discriminator and the binary labels that indicate real or fake samples are randomly flipped during training, thus making the training of discriminator more challenging.

## B MORE QUALITATIVE RESULTS

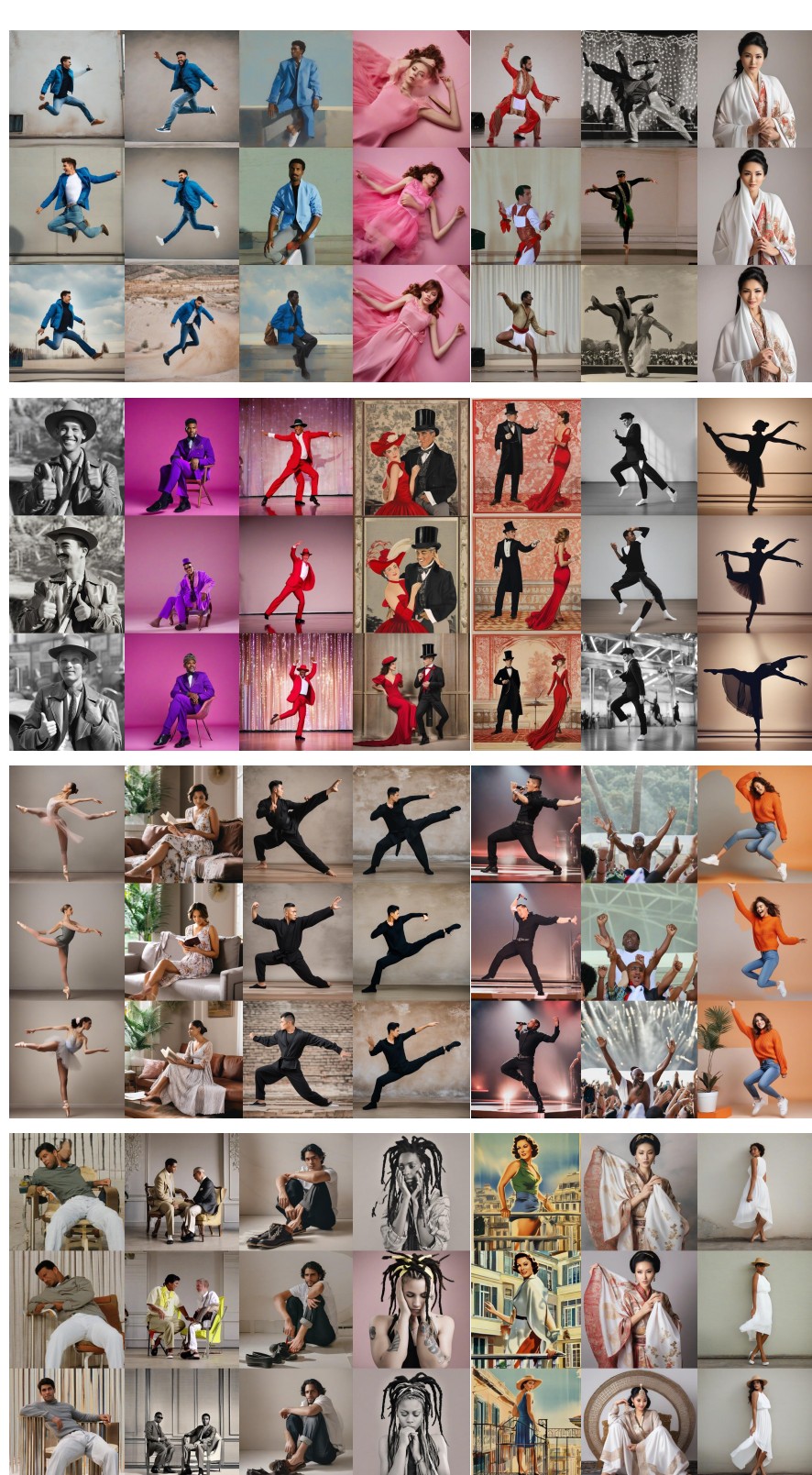

Figure 9: More qualitative comparison for SDXL. Each item contains results generated by pretrained SDXL, naive LoRA and our method from up to bottom.

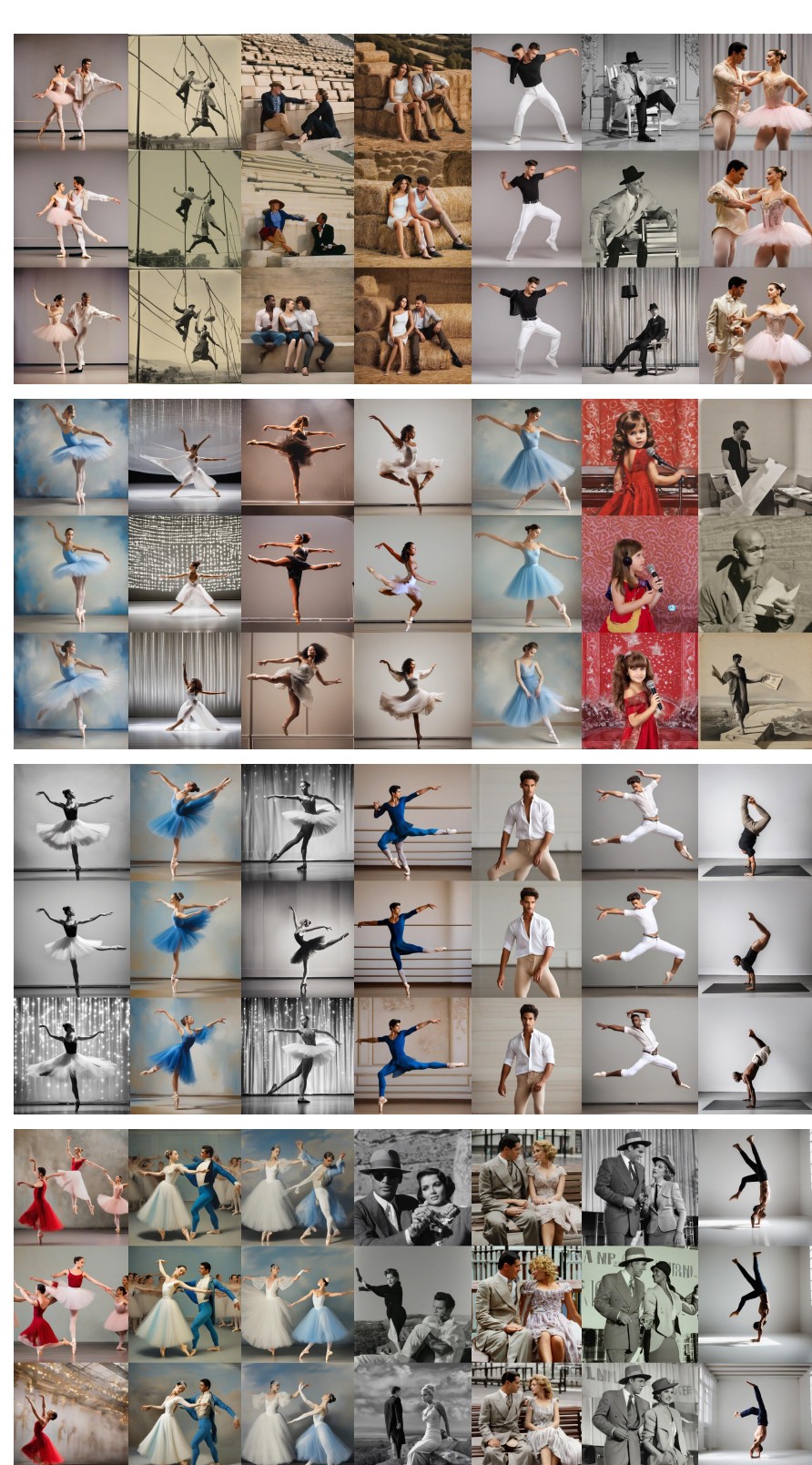

Figure 10: More qualitative comparison for SDXL. Each item contains results generated by pre-trained SDXL, naive LoRA and our method from up to bottom.

