# OpenReview forum: "Enhancing Human Body Generation in Diffusion Models with Dual-Level Prior Knowledge"
_ICLR.cc/2025/Conference — Submitted to ICLR 2025_

### Official Review · Reviewer_ffDo · 2024-10-28

**Soundness:** 2
**Presentation:** 3
**Contribution:** 2
**Rating:** 5
**Confidence:** 4

**Summary:**

This paper proposes to leverage human shape-level and pose-level prior information to guide the diffusion model to generate images with realistic human bodies, with the help of an external human segmentation or pose estimation model.
Several reasonable loss terms are carefully tailored to supervise the attention maps of the target text tokens and an additionally appended pose token.
The authors trained the model on their data and compared the proposed method with the SDXL and LoRA-enhanced SDXL models.
The quantitative results show the model has advantages over its counterparts. The visualization results show the model has enhanced capability in generating anatomically accurate human figures on some manually created prompts with complex semantic meanings.

**Strengths:**

1. The paper is well-written and easy to understand.
2. The overall design of the model and loss function is both straightforward and effective, demonstrating significant improvements in both quantitative and qualitative results.
3. The method addresses key challenges related to human shape and pose in generating realistic human bodies. The proposed loss functions intuitively tackle these target issues.

**Weaknesses:**

1. Some technical details are not explained clearly, such as how $\hat{\mathcal{M}}_{shape}$ is generated from the attention map and what its value distribution is. The criteria for selecting and binding human-related tokens is also unclear.
2. Considering the existence of the HcP approach, the novelty of this paper lies in that the authors additionally use the detected human semantic and pose masks to constrain the attention maps between latent features and specific text tokens. But the spirits of both methods are shared -- supervising the attention maps to align with human semantic or pose masks. In addition, there is no direct comparison with HcP in their results section. This would help readers better understand the incremental improvements or differences of the proposed method.
3. The suppressing, margin, and scaling losses are meaningful. However, some of them may have similar effects. Specifically, the scaling loss might serve similar objectives to those addressed by the suppressing and margin losses. The paper presents only qualitative visual comparisons, lacking quantitative ablation studies to differentiate the individual impact of these loss components.

**Questions:**

1. I wonder why the authors didn't use open-source data like the Human-art dataset as the training set for comparison with the HcP method.
2. How do the authors select human-related tokens to bind regions? Given that adjectives preceding human references (such as "fat" or "thin") and the number of people appearing can affect the generated results, would there be potential issues if we only supervise individual nouns (such as "woman" or "man")?
3. I wonder whether the attention maps $\hat{\mathcal{M}}_{shape}$ have been normalized with the minimum and maximum falling into  [0, 1], as the original attention maps are weighted by a softmax function.
4. What is the evidence of the claims in Line-054? Are the SD-3 and SDXL models trained on large-scale data with enhanced human body image quality?

---

### Official Review · Reviewer_K3hn · 2024-11-03

**Soundness:** 3
**Presentation:** 2
**Contribution:** 2
**Rating:** 3
**Confidence:** 4

**Summary:**

This paper aims to tackle the lack of human body priors in current text-to-image diffusion models. It proposes a two-stage framework that learns pose and shape priors in a way similar to textual inversion and then employs a generative adversarial network to further improve the model. The proposed method is evaluated on a self-collected dataset and outperforms the pretrained and the low-rank adapted SDXL model..

**Strengths:**

\+The quantitative and qualitative comparisons suggest that images generated by the proposed method are better than those by SDXL and SDXL + LoRA.

\+Multiple losses are combined to learn pose and shape priors.

\+The proposed adversarial learning strategy that refines attention maps is reasonable.

**Weaknesses:**

\-The proposed method is not validated thoroughly. Although this paper provides a few visual examples of the ablation study on the proposed components (Fig. 6 - 8), there is no quantitative analysis of this. This paper will become more convincing if more concrete results can be provided.

\-As SD models are not tailored for human images, I believe it is necessary to compare the proposed method with SOTAs that are designed for related tasks. Moreover, the proposed method is inspired by HcP, HyperHuman, and TokenCompose. Even if their codes are not available, it does not mean we cannot implement their key components (or even ControlNet) to investigate the advantages of the proposed method.

\-The experiment setups are not clear and many important details are missing. For example, how many images are used for training/evaluation? What are the shapes for the pose and shape tokens? How to balance different loss terms?

\-The proposed method models body shapes by masks, which are entangled with poses and may fail in cases like loose garments, and inaccurate segmentations.

\- There are many typos in this paper like "the pretrained SD fail" -> "fails", "those human-related token" -> "tokens", "to embed to pose".

**Questions:**

As discussed in the weakness part, my concerns are mainly about the reproducibility and effectiveness of the proposed method. Currently, I think this paper has much room for improvement. I may change my rating after the authors' rebuttal and other reviewers' opinions. Besides, I have the following questions that I wish could be addressed.

Q1: This paper claims even the recent SD3 model struggles with generating human body images. Can more concrete results be provided? Can it benefit from the proposed method as well?

Q2: How is $L_{scale}^{pose}$ defined? Because, unlike shape masks that have insightful meaning, pose masks are merely sticks and points.

---

### Official Review · Reviewer_AX9j · 2024-11-03

**Soundness:** 3
**Presentation:** 2
**Contribution:** 3
**Rating:** 5
**Confidence:** 3

**Summary:**

The paper focuses on human-specific generation using diffusion model, and shows better human generation capabilities than general diffusion models such as SD 1.5 and SDXL, as well as human-specific diffusion model HcP (CVPR24). The paper recognizes the issues in HcP: (1) retention of noisy priors due to the addition between noisy attention map and additional attention maps. (2) limited understanding of human representations beyond the poses. (3) limited generalization power due to supervised learning strategy. To address these limitations, the paper propose a two-stage bind-then-generalize strategy, a composite learning objective, adversarial training to learn real attention map distribution.

**Strengths:**

The proposed method is well-motivated. The paper effectively identifies and analyzes the limitations of HcP, justifying the proposed improvements.

**Weaknesses:**

-	The proposed method is quite complex. While each individual modification to the diffusion model is relatively minor, the model is ultimately a combination of many changes and introduces many additional loss terms. This complexity could make it challenging to balance the losses during training, and the overall approach lacks elegance. The authors should explain how they tune the loss term weights for their result, and the weights for each term. The implementation details in the paper and supplementary do not contain enough details. The authors should consider to provide the specific weights used for each loss term, and an explanation of their process for tuning and balancing these weights during training This would give readers more insight into the relative importance of each component and how to potentially reproduce the results.
-	The comparison is lacking. The paper did not compare to HcP, the method that motivated the paper. It only compared to SD 1.5, SDXL, and LoRA. This is my main concern that leads to the initial rating. Please include a direct comparison with HcP if possible, or a qualitative comparison of the results if quantitative comparison is not possible.

**Questions:**

- The last example from the last row in Figure 6 still seems deformed. I would assume this is an easier case. Why is that? Perhaps the authors can provide an analysis of why their method still struggles with this seemingly simpler case, and discuss what this reveals about potential limitations of their approach
- The dataset for training is crawled from open-source search engines.
  - What specific search terms or criteria were used for crawling?
  - How was the filtering process for "unclear human bodies" defined and implemented?
  - Are there any potential biases in the crawled dataset that might impact the model's performance across different demographics or body types?
  - What advantages does this custom dataset offer over existing human datasets?

**Details Of Ethics Concerns:**

The paper claims that they crawl training images from open-source websites, but it lacks explanation on the exact protocl. I think additional explanation from the authors is needed.

---

### Official Review · Reviewer_urv3 · 2024-11-06

**Soundness:** 2
**Presentation:** 3
**Contribution:** 3
**Rating:** 3
**Confidence:** 4

**Summary:**

This paper introduces a method to improve diffusion models like SD1.5 and SDXL to generate more realistic human bodies and poses. It proposes to increase the richness of human priors used in generation by learning both shape-level and pose-level priors, introduces a new composite objective function, and leverages adversarial training to mimic the real data attention map distribution. The model shows improved quantitative performance over baselines and qualitative analysis is also provided.

**Strengths:**

The paper tackles a clear weakness of existing diffusion models, proposes a number of methods contributions, and shows improved performance in results.

**Weaknesses:**

- My biggest concern is the insufficient comparisons to demonstrate the improvements of the proposed approach over prior work. For example, there is no comparison with previous directly relevant works such as HcP. Although it is noted that HcP is not open-sourced, the authors could use other ways to assess performance of their method vs HcP, such as comparing in the experimental setting of HcP against their reported results. There is also no quantitative ablation studies to demonstrate the importance of the various components and losses of the proposed approach, A few qualitative images is not sufficiently convincing.

- Related, the shape component of the model is not clear to me both in terms of exactly how it works, and its effectiveness (there is no targeted quantitative evaluation of it). Motivations are given such as “factors such as body shape (e.g., thin or heavy) are crucial” and “it remains unclear whether to generate a slender girl or a heavier one”, in Lns. 73-75, but I find it unclear how these scenarios are handled in the shape terms.

- Could enforcing suppression of background region activations lead to decreased quality in generating those parts, when the prompt is not so simple and should carefully control the background as well? It would be useful if the model improves human generation without sacrificing non-human generation so that it can be broadly used. I have some concern if that is the case here or not.

**Questions:**

Could you please elaborate on the shape and background suppression concerns above.

Also, could you comment on the stability of the adversarial training, and the overall training complexity and cost of your method compared to prior works.

---

### Official Review · Reviewer_8ah6 · 2024-11-06

**Soundness:** 2
**Presentation:** 3
**Contribution:** 2
**Rating:** 3
**Confidence:** 4

**Summary:**

The paper addresses the problem of anatomical errors in generating human body images using diffusion models. The proposed solution is a dual-level prior knowledge injection method that enhances pretrained diffusion models by learning both shape and pose level human priors. This is implemented with a two-stage training strategy, including “semantic attention binding” and “adversarial attention rectification”. Specifically, they add a learnable pose-aware token to text prompts and use multiple loss functions and adversarial training to refine cross-attention maps, leading to more accurate human body shapes demonstrated mainly through qualitative results.

**Strengths:**

1. The paper addresses a issue of significance, as diffusion models are often observed to struggle with generating accurate anatomical structures.
2. Incorporation of pose and shape priors is intuitive.
3. The quantitative and qualitative results are impressive in terms of fidelity.

**Weaknesses:**

1. There is no quantitative evaluation of the anatomical plausibility of generated human bodies, and the qualitative results still reveal issues (e.g., Fig. 9 shows disproportionate bodies in the 1st column, overlapping bodies in the 2nd, and anomalies like extra hands and hands on feet in the 3rd, ...). Overall, I could not assess if the proposed method fully or partially addresses the problem. To strengthen this evaluation, I suggest deriving a metric for anatomical plausibility / correctness based on current pose estimation tools such as [1, 2].

2. The ablation study (Fig. 7) suggests that the method only succeeds when both pose and shape priors are used. However, the results are not particularly intuitive in demonstrating how each prior individually contributes, as all three rows appear to produce similar body shapes. Additional results or explanations are appreciated in order to understand how these priors function.

[1] Güler, Rıza Alp, Natalia Neverova, and Iasonas Kokkinos. "Densepose: Dense human pose estimation in the wild." Proceedings of the IEEE conference on computer vision and pattern recognition. 2018.

[2] Zhu, Wentao, et al. "Motionbert: A unified perspective on learning human motion representations." Proceedings of the IEEE/CVF International Conference on Computer Vision. 2023.

**Questions:**

An alternative solution to the problem is to implement a two-step framework where the pose and shape is generated first [1, 2], followed by rendering the image with a diffusion model conditioned on the pose [3, 4].
Could you clarify the motivation for choosing this particular method? In what scenarios would this solution be preferable to the alternative approach I mention.

[1] Loper, Matthew, et al. "SMPL: A skinned multi-person linear model." Seminal Graphics Papers: Pushing the Boundaries, Volume 2. 2023. 851-866.

[2] Shafir, Yonatan, et al. "Human motion diffusion as a generative prior." arXiv preprint arXiv:2303.01418 (2023).

[3] Zhang, Lvmin, Anyi Rao, and Maneesh Agrawala. "Adding conditional control to text-to-image diffusion models." Proceedings of the IEEE/CVF International Conference on Computer Vision. 2023.

[4] Ju, Xuan, et al. "Humansd: A native skeleton-guided diffusion model for human image generation." Proceedings of the IEEE/CVF International Conference on Computer Vision. 2023.

---

### Meta-Review · Area_Chair_zgis · 2024-12-19

**Metareview:**

This paper proposes an approach for improving realistic human body generation of diffusion models with dual-level prior knowledge. 5 reviewers consistently give negative scores with major concerns in method design and experimental validation. The authors did not provide rebuttal. Therefore, the area chair would recommend rejecting this paper and suggest the authors to improve the paper and presentation for future submissions.

**Additional Comments On Reviewer Discussion:**

There was no rebuttal, so no discussions.

---

### Decision · Program_Chairs · 2025-01-22

Reject